# Screening of Novel Laccase Producers—Isolation and Characterization of Cold-Adapted Laccase from *Kabatiella bupleuri* G3 Capable of Synthetic Dye Decolorization

**DOI:** 10.3390/biom11060828

**Published:** 2021-06-02

**Authors:** Katarzyna M. Wiśniewska, Aleksandra Twarda-Clapa, Aneta M. Białkowska

**Affiliations:** Institute of Molecular and Industrial Biotechnology, Lodz University of Technology, Stefanowskiego 4/10, 90-924 Łódź, Poland; kma.wisniewska@gmail.com (K.M.W.); aleksandra.twarda-clapa@p.lodz.pl (A.T.-C.)

**Keywords:** laccase, psychrophilic, *Kabatiella bupleuri*, decolorization, synthetic dyes

## Abstract

Psychrophilic laccases catalyzing the bond formation in mild, environmentally friendly conditions are one of the biocatalysts at the focus of green chemistry. Screening of 41 cold-adapted strains of yeast and yeast-like fungi revealed a new laccase-producing strain, which was identified as *Kabatiella bupleuri* G3 IBMiP according to the morphological characteristics and analysis of sequences of the D1/D2 regions of 26S rDNA domain and the ITS1–5,8S–ITS2 region. The extracellular activity of laccase in reaction with 2,2’-azino-bis(3-ethylbenzothiazoline-6-sulfonic acid) (ABTS) at the optimal pH 3.5 was 215 U/L after 15 days of growth in a medium with waste material and 126 U/L after 25 days of cultivation in a defined medium. Copper (II) ions (0.4 mM), Tween 80 (1.0 mM) and ascorbic acid (5.0 mM) increased the production of laccase. The optimum temperature for enzyme operation is in the range of 30–40 °C and retains over 60% of the maximum activity at 10 °C. New laccase shows high thermolability—half-life at 40 °C was only 60 min. Enzyme degradation of synthetic dyes was the highest for crystal violet, i.e., 48.6% after 1-h reaction with ABTS as a mediator. Outcomes of this study present the *K. bupleuri* laccase as a potential psychrozyme for environmental and industrial applications.

## 1. Introduction

Laccase is an enzyme belonging to the group of multicopper oxidases and catalyzing the oxidation in which a substrate loses a single electron and is converted to a free radical. The resulting unstable radical may further undergo laccase-catalyzed oxidation or non-enzymatic reactions such as hydration, disproportionation and polymerization. In laccase-catalyzed reactions, diphenols undergo 4-electron oxidation. During this reaction, the Cu(II) ion is reduced to the Cu(I) ion. Moreover, Cu(I) reduces molecular oxygen (O_2_) to produce two water molecules, and is oxidized back to Cu(II) (Figure 1) [1]. Laccases display a wide substrate specificity, including phenols, phenolic compounds, aminophenols, aromatic amines, lignins and steroid hormones. For this reason, they are used in many different industries, such as paper, petrochemical, food, and textile [2]. The use of laccases adapted to low temperatures is particularly interesting. The textile, paper, printing and tannery industries produce huge amounts of waste containing synthetic dyes toxic to the environment. It is estimated that about 5–10% of them could remain persistent in wastewater. The use of laccases allows carrying out the processes of their degradation without using chemical and physical methods [3,4,5]. Laccase can also be used in the processes of detoxification of environments polluted with polycyclic aromatic compounds [6,7].

Laccases are products of secondary metabolism and their synthesis is influenced by limiting factors, such as the type of culture, its duration and the composition of the medium, or by the addition of inducers, as the genes encoding laccases contain regulatory sites, e.g., metal responsive elements (MREs) and xenobiotic responsive elements (XREs) [9]. The most common inducers include Cu(II) ions, aromatic compounds (veratryl alcohol, vanillic acid, 2,5-xylidine, ferulic acid, syringaldazine, guaiacol, p-hydroxybenzoic acid, 3,4-dimethoxycinnamic acid, 4-hydroxybenzaldehyde, vanillin), vitamins and essential amino acids (tryptophan, methionine, glycine, valine, biotin, riboflavin), organic solvents (ethanol), xenobiotics (4-n-nonylophenol, aniline, oxidized derivatives of diquat and N,N′-dimethyl-N-(5-chloro-4-hydroxyphenyl), urea, surfactants (Tween 80) and ethidium bromide [10].

Laccases are ubiquitous enzymes in nature. They were isolated from higher plants, insects and bacteria, but their main source is white rot fungi. The first plant laccase was isolated from a Japanese tree, *Rhus vernicifera* [11]. Fungal laccases were isolated later, in 1896, by Bertrand [12]. Among the fungi producing this enzyme we can distinguish *Trametes versicolor*, *Trametes ochracea*, *Trametes hirsuta*, *Cerrena maxima* and *Coriolopsis polyzona*, and among bacteria—*Escherichia coli*, *Bacillus subtilis*, *Streptomyces antibioticus* and *Azospirillum lipoferum* [9,13].

In this study we conducted a screening of psychrophilic and cold-adapted yeast and yeast-like fungi for the production of laccases working at low temperatures. Decreasing the temperature enables lowering of the costs of energy needed for heating the waste—on the contrary, the process can be conducted at the site of contamination. Despite the undoubted advantages of low-temperature-adapted enzymes, little information is available on psychrophilic laccases. Rovati et al. [14] noted the laccase activity at 15 °C from 25 yeast strains isolated from soil samples from de Mayo/King George Island. Certain adaptations to low temperatures were also observed for the laccase from *Cryptococcus albidus* (renamed to *Naganishia albida*) [15] and from *Colletotrichum lagenarium* [16].

The characterization of the *Kabatiella bupleuri* laccase conducted in this study showed that it is an enzyme of interest from a biotechnological point of view and that it can be used in in situ decolorization of synthetic dyes.

## 2. Materials and Methods

### 2.1. Organism Selection and Culture Conditions

To search for producers of cold-adapted laccases, 41 strains of psychrophilic yeast and yeast-like fungi were used. Furthermore, 17 strains were purchased from the Italian collection, Industrial Yeasts Collection DBVPG (Perugia, Italy). The remaining 26 strains were isolated from soil samples located at 10 different sites in the vicinity of the Henryk Arctowski Polish Antarctic Station on King George Island (62°09′41″ S, 58°28′10″ W) and are a part of the collection of Antarctic microorganisms in the Institute of Molecular and Industrial Biotechnology, Lodz University of Technology (Stefanowskiego 2/22, 90-537, Lodz, Poland) [17]. Cultivation included the following stages:

Plate screening on the selection media containing the inducers of laccase production:

(1) LBM medium supplemented with ABTS (2,2′-azino-bis(3-ethylbenzothiazoline-6-sulfonic acid)) (in g/L: 4.0 glucose, 4.0 KH_2_PO_4_, 0.5 ammonium tartrate, 0.5 MgSO_4_ × 7H_2_O, 0.01 CaCl_2_ × 2H_2_O, 0.01 yeast extract, 0.001 CuSO_4_ × 5H_2_O, 0.001 Fe_2_(SO_4_)_3_, 0.001 MnSO_4_ × H_2_O, 1.0 ABTS, 16.0 agar, pH 5.5–6.0), and (2) medium with guaiacol (in g/L: 10.0 glucose, 3.0 peptone, 0.6 KH_2_PO_4_, 0.6 K_2_HPO_4_, 0.5 MgSO_4_, 0.05 MnSO_4_ × H_2_O, 0.001 ZnSO_4_, 0.0005 FeSO_4_, 0.2 guaiacol, 20.0 agar, pH 6.0).

Inoculated plates with selective media were incubated at 20 °C for 20 d. The development of an intense bluish-green and reddish-brown color around the wells on ABTS or guaiacol substrate, respectively, was considered as a positive test for laccase activity.


Submerged culture of laccase producers selected during plate screening on the following media: (1) Olga medium [18], (2) modified Kirk medium (LMM) [19], (3) Sivakumar medium [20] and (4) mineral medium in order to select the most optimal medium for laccase production by the cold-adapted fungi. (1) The Olga medium contained (in g/L): 3.0 peptone, 10.0 glucose, 0.6 KH_2_PO_4_, 0.001 ZnSO_4_, 0.4 K_2_HPO_4_, 0.0005 FeSO_4_, 0.05 MnSO_4_ and 0.5 MgSO_4_ (pH 6.0); (2) LMM contained (in g/L): 10.0 glucose, 1.0 yeast extract, 2.0 ammonium tartrate, 1.0 KH_2_PO_4_, 0.5 MgSO_4_ × 7H_2_O, 0.5 KCl and 0.15 mM CuSO_4_ × 5H_2_O (pH 5.5–6.0); (3) Sivakumar medium contained (in g/L): 20.0 soluble starch, 2.5 yeast extract, 1.0 KH_2_PO_4_, 0.05 Na_2_HPO_4_, 0.5 MgSO_4_, 0.01 CaCl_2_, 0.01 FeSO_4_, 0.001 MnSO_4_, 0.001 ZnSO_4_ and 0.002 CuSO_4_ (pH 5.5); and (4) mineral medium contained (in g/L): 2.0 glucose, 1.0 (NH_4_)_2_SO_4_, 1.0 K_2_HPO_4_, 0.5 MgSO_4_ × 7H_2_O, 0.5 NaCl, 0.01 MnSO_4_, 0.001 CuSO_4_ × 5H_2_O (pH 5.5–6.0).Submerged culture of *Kabatiella bupleuri* strain in the Sivakumar medium with the addition of several potential inducers of laccase biosynthesis (concentrations specified in the Results) such as copper (II) ions, Tween 20, Tween 80, ABTS, guaiacol, syringaldazine, ethidium bromide, catechol, 2,5-xylidine, veratryl alcohol, vanillin and ascorbic acid, in order to find the inducer that causes the highest enzyme activity (U/L).Submerged culture in the modified Sivakumar medium with the selected inducers (copper (II) ions and Tween 80), in which the starch was replaced by waste products in concentrations of 20 and 100 g/L from the agri-food and brewing industry and compounds rich in lignocellulose (carrot pomace, apple pomace, potato pulp, spent grain from the brewery, lignin, rye straw, straw briquette).


The strains were pre-cultured in the rich medium containing 5.0 g/L yeast extract and 10.0 g/L peptone at 20 °C for 3 days. This pre-culture (2.6 mL) was used to inoculate 65 mL of different laccase producing media in a 250 mL Erlenmeyer flask. Cultures were conducted for 25–30 days at 20 °C with shaking (150 rpm). Inducers were added after 3 days of growth.

### 2.2. Laccase Extraction from the Biomass

Fungi biomass harvested after submerged cultures was disrupted by shaking with glass beads. The wet biomass was disintegrated in 20 mM Tris-HCl buffer pH 7.4, enriched with 0.1% Triton-X100, 100 mM KCl, 8 mM MgCl_2_, 150 mM NaCl and 1 mM PMSF. The sample was vortexed for 30 s and left on ice for 30 s. The procedure was repeated 12 times. The sample was centrifuged at 1500× *g* for 5 min at 10 °C and the insoluble cell debris was discarded.

### 2.3. Genetic Identification of Laccase-Producing Strain

#### 2.3.1. DNA Extraction

Yeast and yeast-like strains were cultivated in the flasks with medium containing 20.0 g/L glucose, 10.0 g/L yeast extract and 20.0 g/L peptone at 20 °C and 150 rpm. Biomass was centrifuged (5000× *g*, 10 min, 4 °C) and frozen at −80 °C. Next, 100 g of frozen biomass was ground in the mortar with the liquid nitrogen to a uniform powder and transferred to a 15 mL Falcon tube. One milliliter of the extraction buffer (200 mM Tris-HCl, pH 7.5; 25 mM EDTA, pH 8.0; 0.5% SDS; 250 mM NaCl) and 2 mL of the phenol:chloroform:isoamyl alcohol (25:24:1, *v*:*v*:*v*) mixture was added; the tube was gently shaken for 10 min at the room temperature and centrifuged at 11,000 rpm for 30 min. Water phase was collected, supplemented with 5 µL 10 mg/mL RNAse (EURx, Gdansk, Poland), incubated for 1 h at 37 °C, and extracted (equal-volume) with phenol:chloroform:isoamyl alcohol (25:24:1, *v*:*v*:*v*) with vigorous shaking. Water phase was collected after centrifugation (13,000 rpm, 15 min, 4 °C). DNA was precipitated by addition of 0.1 volume of 3 M sodium acetate pH 5.2 and 0.6 volume of ice-cold isopropanol. The sample was incubated over night at 20 °C and centrifuged (15,000 rpm, 15 min, 4 °C). DNA precipitate was washed twice with 70% ethanol (150 µL), centrifuged and dried for 2–3 min. DNA was dissolved in 60 µL of sterile RNAse-free distilled water.

#### 2.3.2. PCR Amplification of D1/D2 and ITS1–5,8S–ITS2 Regions

PCR amplification of the D1/D2 regions of 26S rDNA domain and ITS1–5,8S–ITS2 region were performed using primers LR6/ITS5 and RLR3/V9 (Table 1), respectively, with genomic DNA as a template. The reaction condition of PCR was an initial denaturation step at 98 °C for 3 min; 40 cycles at 98 °C for 30 s denaturing, annealing at 44 °C for 30 s and extension at 72 °C for 48 s; a final extension of 72 °C for 5 min followed by maintenance at 4 °C. PCR products were purified using a GeneMatrix Basic DNA Purification Kit (EURx, Gdansk, Poland) and sequenced using starters NL1FWD/NL4REV for region D1/D2 and ITS1/ITS4 for fragment ITS1–ITS2 (Table 1).

#### 2.3.3. Data Analysis

Sequences of region D1/D2 and fragment ITS1–ITS2 were analyzed in Nucleic Acid Sequence Massager (http://www.cmbn.no/tonjum/seqMassager-saf.htm) (accessed on 1 June 2017) and compared to the sequences in GenBank at NCBI (http://www.ncbi.nlm.nig.gov/) (accessed on 27 April 2021) using the BLASTn algorithm at NCBI. The phylogenetic trees were calculated by the fast minimum evolution method using the results obtained using BLAST pairwise alignments.

### 2.4. Raw Materials

Apple pomace, carrot pomace, potato pulp, rye straw and brewer’s spent grain (BSG) (approx. 80% moisture content) were dried at 60 °C to 90% dry matter to be stored safely. Lignin (Matocell, TZMO SA, Torun, Poland) was crushed to an average particle size of about 1 cm. Straw briquette (Eko-Pellets, Nowy Tomysl, Poland) was in the form of light, straw-colored cylindrical particles with diameter about 6–8 mm and length about 5–50 mm. Materials were sterilized at 121 °C for 15 min before being used in culture.

Chemical characterization of BSG including estimation of cellulose, hemicellulose, lignin, crude protein content and ash content was determined according to Meneses et al. [21].

### 2.5. Purification of the Laccase by Ammonium Sulphate Fractioning

Laccase G3 was extracted by the method of fractional desalting. Post-culture liquid was centrifuged and ammonium sulphate was added in portions until the final concentration of 40% to the supernatant placed on ice. After 1-h incubation and centrifugation (8000× *g*, 10 min), another portion was added to the supernatant until final concentration of 60%. The process was repeated twice more to obtain 80% and 100% saturation with ammonium sulphate. The pellet containing precipitated laccase was recovered by centrifugation at 8000× *g* for 10 min at 4 °C and then dissolved in 10 mM acetate buffer pH 5.5. The pre-purified laccase extract was dialyzed for 24 h against water and then analyzed for its activity and protein concentration. Samples were concentrated prior to SDS-PAGE analysis in the Amicon Ultra-15 centrifugal filter units with MWCO of 10 kDa (Merck Millipore, Burlington, MA, USA).

### 2.6. SDS-PAGE

Molecular mass of the laccases was determined by SDS-PAGE [22] in 12% polyacrylamide gel and 0.1 M Tris-glycine buffer pH 8.3. Samples were denatured for 10 min at 96 °C in a sample buffer with 1% SDS and 10% β-mercaptoethanol. Precision Plus Protein Unstained Standards (Bio-Rad, Hercules, CA, USA) were used as mass standards. Bands were visualized with Bio-Safe Coomassie Stain (BioRad, Hercules, CA, USA). Molecular mass was determined using Quantity One software (BioRad, Hercules, CA, USA) on densitometer BioRad GS-800.

### 2.7. Effect of Temperature and pH on Laccase Activity and Stability

The optimum temperature for laccase activity was determined by incubating the reaction mixture over a temperature range of 0–80 °C at pH 3.5. To investigate the thermostability of the enzyme, it was incubated in the range of 5–80 °C for 0–120 min within a 30 min interval, followed by measuring the remaining enzyme activity at pH 3.5. The effect of pH on laccase activity and stability was determined using 10 mM Britton–Robinson buffer for pH 2.0–10.0, at the optimum temperature. The enzyme was incubated for 24 h and then the relative activity of laccase was determined. All reactions were performed in triplicates.

### 2.8. Substrate Specificity of Laccase

Substrate specificity of laccase from *K*. *bupleuri* was estimated using ABTS and syringaldazine (ε_526_ = 6.5 × 10^4^ M^−1^ cm^−1^; conc. 5.5–40.0 µM) as a substrate. Firstly, optimum pH for syringaldazine as obtained in 10 mM Britton–Robinson buffer pH 2.0–10.0 and with 30 µM concentration. Activity determination was carried out at 30 °C. Kinetic parameters, namely, K_m_ and V_max_, were determined using the Lineweaver–Burk plots.

### 2.9. Decolorization of Synthetic Dyes

Four synthetic dyes: methylene blue (λ = 668 nm), basic fuchsine (λ = 544 nm), Coomassie Brilliant Blue (λ = 555 nm) and crystal violet (λ = 590 nm) were decolorized with the purified laccase. Stock solutions of dyes were prepared in sterile distilled water. The total volume of the reaction mixture was 2 mL, which contained acetate buffer (100 mM, pH 3.5), synthetic dye (250 mg/L), 200 µL of purified laccase (0.1 U) and 1.0 mM of ABTS as a redox mediator. The mixture was incubated at 30 °C for 60 min, and then the absorbance was measured. The percentage was determined spectrophotometrically as the relative decrease in absorbance at each maximal absorbance wavelength of the dyes. The decolorization of dye, expressed as dye decolorization (%) was calculated by means of Formula (1):decolorization (%) = [(A_i_ − A_f_)/A_i_] × 100,(1)
where: A_i_—initial concentration of the dye and A_f_—final concentration of the dye. All reactions were performed in triplicates.

### 2.10. Analytical Methods

Laccase activity was determined by the oxidation of ABTS method [23]. Biomass was discarded (5000× *g*, 15 min, 4 °C) and the laccase activity in the supernatant was determined. The non-phenolic dye ABTS is oxidized by laccase to the more stable and preferred state of the cation radical. Oxidation of ABTS was monitored by determining the increase in A_420_ (ε_420_, 3.6 × 10^4^ M/cm). The reaction mixture contained 1 mM substrate (ABTS), 150 µL of 0.05 M sodium acetate buffer pH 3.5, and 50 µL of culture supernatant, and was incubated for 10 min. Absorbance was read at 420 nm in a spectrophotometer against a suitable blank with water instead of the enzyme. One unit (U) was defined as the amount of the laccase that oxidized 1 μmol of ABTS substrate per min.

Lignin peroxidase, manganese-dependent peroxidase and manganese-independent peroxidase activity was determined according to Pointing and coworkers [24].

Protein concentration was determined according to Bradford [25] using BSA as a standard [25].

### 2.11. Statistical Analysis

Calculations were done using Microsoft Excel version 2007. Experimental values were reported as the means ± s.e. Statistical comparisons were made using a one-way ANOVA (Tukey’s test, *p* < 0.05). All calculations of statistical significance were made using the Graph Pad Prism5 and SPSS ver.11. Graphs were plotted using SigmaPlot version 11 and Graph Pad Prism5.

## 3. Results and Discussion

### 3.1. Screening of Yeast and Yeast-Like Fungi for Laccase Activity

The search for a new psychrophilic laccase began with screening of yeast and yeast-like fungi strains on the selection media containing ABTS or guaiacol as indicators of extracellular activity of the enzyme. The plates were incubated for 20 days at 20 °C. A green ring around the colonies on the ABTS medium or a brown one on the guaiacol medium was observed for four strains, i.e., *Cryptococcus gastricus* DBVPG, *Cryptococcus aerius* DBVPG, *Leuconeurospora* sp. D59 IBMiP and strain G3 IBMiP (Figure 2). These strains were then cultivated in Sivakumar, Olga, Kirk and mineral media for 25 days at 20 °C with 200 rpm shaking. For the first three of them, both the extracellular and intracellular activity of laccase was very low and did not exceed 1 U/L. Much higher activity (~35 U/L), was determined for the G3 strain IBMiP (Figure 3).

H_2_O_2_ did not alter the oxidation of ABTS, indicating the lack of peroxidase activity. Lignin peroxidase and manganese peroxidase were not detected either.

The growth dynamics of the G3 strain IBMiP indicated that the extracellular laccase is a secondary metabolite. Its maximum accumulation in the supernatant takes place in the stationary growth phase of the strain and on day 20 it reaches the value of 35.21 ± 0.64 U/L. Protein concentration in the supernatant increases steadily during the cultivation, regardless of G3 laccase biosynthesis, and on day 20 it is at the level of approx. 46 mg/L (Figure 3).

### 3.2. Genetic Identification of Strain G3 IBMiP

In order to determine the taxonomic identification of the G3 IBMiP strain, genomic DNA was isolated and the fragments encoding the D1/D2 region of the large 26S rDNA subunit and the ITS1–5,8S–ITS2 region between the genes encoding the 18S and 26S rDNA subunits were amplified. Purified DNA fragments were sequenced and aligned using BLASTn with the homologous sequences deposited in GenBank. The obtained amplicon sequences were deposited into GenBank (accession numbers MW450856.1 and MW450864.1). The coding sequence for the D1/D2 region showed more than 99% similarity to that of the *Kabatiella bupleuri* strain F278240 (JN886792.1), while for the sequence of the ITS1 and ITS2 region, the greatest similarity was observed for *K. bupleuri* CBS 131304 (NR_121524.1). The morphological, physiological and biochemical characteristics of the strain confirmed its belonging to the species *K. bupleuri* (data not shown).

Phylogenetic trees were prepared to investigate the similarity of the strain G3 IBMiP to other yeast (Appendix A). The results revealed a high similarity of *K. bupleuri* to the genera *Aureobasidium* or *Hormonema*, to which they were originally classified. However, molecular analysis of rRNA-encoding genes by Bills et al. [26] allowed for identification of the new species. Further research carried out in the USA and Thailand confirmed the differences between the genera *Aureobasidium* and *Kabatiella* in terms of different growth conditions [27].

### 3.3. Optimization of Growth Conditions of K. bupleuri to Increase Laccase Biosynthesis

#### Inducers for Laccase Synthesis

In order to facilitate the extracellular activity of *K. bupleuri* laccase G3 in the previously selected Sivakumar medium, the concentration of copper (II) ions was increased 40- and 100-fold compared to the starting medium (0.01 mM Cu^2+^; the values were selected based on the literature data [28,29]). Copper (II) ions are considered to be an efficient inducer of laccase biosynthesis as they are part of the active site by being involved in the transfer of electrons. Most monomeric laccases contain four copper atoms at the following sites: the type I copper (T1), type II copper (T2) and two of type III copper (T3). The T2 and T3 copper sites form a trinuclear center that is involved in the catalytic mechanism of the enzyme [1].

The addition of copper (II) ions to the medium significantly increased the laccase activity in the supernatant by approx. 50% for 20 days (Figure 4). The highest activity, about 50 U/L, was recorded after 24 days of cultivation in a medium with 0.4 mM CuSO_4_. Similar activity values were observed for the higher concentration of copper ions. Therefore, taking into account the economic advantages of the process, a lower concentration was used in further studies for the biosynthesis of laccase. It has been demonstrated that copper (II) ions are an effective inducer of laccase production at concentrations of 0.15–5.0 mM [7,28,29,30,31,32,33,34,35,36]. Chen et al. described a significant role of this enhancer in regulating the expression of genes encoding laccases in *Volvariella volvacea* V14 [31]. By gradually increasing the concentration of Cu^2+^ to 0.2 mM, an increase in the amount of RNA was observed. At the higher concentration (0.3 mM) the effect was opposite and a 70% decrease in the amount of transcript was observed. Guo et al. observed 30× higher activity of an extracellular laccase at 1.0 mM Cu^2+^ [37]. The production of the enzyme was also facilitated by these ions at concentrations of 1.5–2.0 mM for *Trametes pubescens* MB89. The authors have also demonstrated the impact of time of the addition of the ions to the medium—addition at day 4 was the most favorable [38]. Both Galhaup and Haltrich [38] and Klonowska et al. [30] observed that the addition of copper (II) ions to the sterile medium partially inhibited the growth of fungal biomass and, consequently, the amount of laccase produced. A similar effect was noted with too-late dosing of these ions to the substrate. This was also confirmed in the present study, where the best effect of induction of G3 laccase production by the *K. bupleuri* strain was observed when copper (II) ions were added to the medium after 3 days of growth. In the case of laccase produced by *Monilinia fructicola*, it was the day 2 of culture. In the presence of 1 mM Cu^2+^ and Mg^2+^ ions, a 5-fold increase in the activity of this enzyme was observed in the post-culture liquid compared to the culture without inducers [39].

The obtained level of *K. bupleuri* laccase activity after induction with copper (II) ions was still too low to be able to test the potential of the selected oxidoreductase. Therefore, the medium with the addition of the ions was subjected to further modifications and enriched with various groups of compounds for which an activating effect on laccase biosynthesis was shown: (1) surfactants (Tween 20, Tween 80), (2) electron mediators in oxidoreductive reactions (ABTS, guaiacol, syringaldazine), (3) compounds structurally similar to substances that naturally induce the synthesis of ligninolytic enzymes (catechol, 2,5-xylidine, veratryl alcohol, vanillin), and ethidium bromide. Each of these compounds was added to the medium in various concentration ranges (data not shown) and their effect on G3 laccase activity was examined. Figure 5 and Appendix A show the optimal inducer concentrations at which the maximum level of extracellular enzyme activity was reached.

A clear positive effect on the activity of laccase in *K. bupleuri* culture was found for Tween 80. After 20 days of cultivation in a medium enriched with 0.4 mM Cu^2+^ and 1 mM Tween 80, over 50% increase in activity to the level of about 70 U/L was observed. This non-ionic surfactant is added to the medium, e.g., to increase the secretion of proteins [40,41]. Similar results were obtained for other fungal laccases from *Marasmius quercophilus* [30], *Pholiota* sp., *Peniphora* sp. and *Peniophora* sp. [42]. The impact of Tween 80 was also investigated by Pointing et al., however, it was less significant than 2,5-xylidine as an inducer for the production of laccase by *Pycnoporus sanguineus* CY788 [24]. In turn, for the optimal synthesis of laccase by *P. sanguineus* G05, a medium containing Tween 80 and 2,5-xylidine simultaneously was tested, resulting in an over 6-fold increase in the enzyme activity [43].

A significant positive effect on the activity of *K. bupleuri* laccase was also observed for ethidium bromide, veratryl alcohol, syringaldazine and catechol. These compounds increased their activity by approx. 10–20% compared to the control sample containing only 0.4 mM Cu^2+^. Numerous authors have reported an increase in laccase activity as a result of the addition of veratryl alcohol to the culture of *Botryosphaeria* sp. [44,45], *Phlebia brevispora*, *Phlebia radiata*, *Daedalea flavida*, *Polyporus sanguineus* [46], *T. versicolor* [34] and bacterium *Chromohalobacter salexigens* [47], syringaldazine—to the culture of *Coriolus hirsutus* [48], ethidium bromide—to the culture of *Cyathus bulleri* [49], and finally catechol—to the culture of *T. versicolor* [50]. Catechol was added to the culture at 72 h of cultivation and after another 24 h, which yielded a 10-fold increase in activity to the level of approx. 1 U/mL. Lower (only 10% higher laccase activity, than in the control) enhancing ability for catechol was observed by Galhaup and Haltrich [38]. Inducers often reported as effective for laccases, such as 2,5-xylidine [31,38,47,51,52], guaiacol [33,38,46,47,48], ABTS [34] and vanillin [53] were found ineffective for the G3 laccase and in the research of Guo et al. [37]. Other fungal laccase inducers, however not tested in this study, were reported in the literature: D-phenylalanine [54], phenylhydrazine, 4-methylcatechol [33], organic solvents like ethanol [55,56,57] or acetone [58], 4-n-nonylphenol [59], actinomycin D [60], ferulic acid [57] and 3,4-dimethoxycinnamic acid [61].

### 3.4. Effect of Ascorbic Acid on Laccase Production

*K. bupleuri* yeast-like fungi grown in Sivakumar medium produces carotenoid pigments, which causes the biomass to become orange to dark red in color. The production of extracellular *K. bupleuri* laccase is closely related to the appearance of the amber and, with time, dark brown color of the culture liquid. On completion of the culture, the pigment is observed in the biomass and supernatant. It has been observed that the intensity of its color is correlated with the activity of laccase; the more distinct the color of the post-culture liquid, the higher the activity of the enzyme in the supernatant. This may be due to the fact that laccase-inducing compounds (see Figure 5) increase the biosynthesis level of other oxidoreductases, including, for example, peroxidases, which catalyze the polymerization reactions of carotenoid dye molecules, leading to the formation of aggregates with increased color saturation. Thurston pointed out that in a number of fungi, laccase activity is associated with pigment formation and it is certainly possible that this is its natural role in *Aureobasidium pullulans* which is closely related to *K. bupleuri* [62]. The highest activity of laccase was observed for strains producing dark purple or vinaceous pigment. Colorless strains, despite growing on media containing lignin as the only carbon source, showed no laccase activity [52].

The presence of pigment in the supernatant makes further steps of purifying *K. bupleuri* G3 laccase much more difficult. Therefore, attempts were made to inhibit the dye polymerization process by adding ascorbic acid to the culture medium—an antioxidant compound that increases the stability of carotenoids [63]. As a result of optimization of the concentration of this additive in terms of laccase biosynthesis, it was shown that its supplementation at a concentration of 3.0 mM and 5.0 mM caused a significant increase in the activity of laccase, respectively, by 50 and 100% after 25 d compared to the control sample (Figure 6), while reducing the intensity of the color of the supernatant. It is presumed that the reduced polymerization of the dye molecules due to the presence of the strong antioxidant made it possible to increase the pool of a free enzyme with the active site readily available for the substrate. Moreover, the addition of ascorbic acid, significantly reducing the color of the supernatant, improved the quality of its filtration and facilitated further stages of enzyme purification.

### 3.5. Waste Materials in Laccase Synthesis

Taking into account the global pursuit of rational waste management and analyzing the economic considerations of the laccase biosynthesis process, the *K. bupleuri* G3 strain was cultured in the Sivakumar medium with starch replaced by waste products from the agri-food and brewing industries, such as BSG, carrot pomace, apple pomace, potato pomace, lignins, rye straw and straw briquettes (Figure 7). Their selection was dictated by the high content of lignins, which may be inducers of G3 laccase biosynthesis (Figure 7b). Originally, to select the best waste carbon source, two concentrations were used: 20 and 100 g/L. The waste material was the only carbon source in the medium. The first concentration, regardless of the type of waste, turned out to be too low for efficient growth of *K. bupleuri* and extracellular laccase biosynthesis (data not shown). For the latter, the highest enzyme activities were obtained using spent grains from the brewery, BSG (Figure 7a), and the activity levels were similar to those achieved with the starch medium.

An important advantage of laccase biosynthesis using BSG as the only carbon source in the medium was a significant reduction in the cultivation time to reach the maximum enzyme activity (130 U/L)—from 20 to 25, depending on the type of substrate, to 14 days. In the following days, there was a sharp decrease in the activity of laccase in the supernatant, despite the constant increase in the protein concentration (reaching 1.16 g/L after 22 days) (Figure 7c). The obtained protein concentrations in the media containing waste products as a carbon source are clearly higher than those determined after culturing in defined media (maximum 30–50 mg/L). This is due to the fact that most of the used plant waste materials are themselves rich in proteins. In further studies, incubation of the uninoculated medium was carried out in parallel to the proper culture to eliminate the protein content from the waste in determination of concentration.

To simplify and minimize substrate costs for efficient *K. bupleuri* laccase biosynthesis, all components of the modified Sivakumar medium (except for spent grains) were replaced with water. This procedure was dictated by the chemical composition of this waste material (Figure 7b). It is rich in hemicellulose, cellulose, lignin and proteins—substances that are a good source of carbon, nitrogen and laccase biosynthesis inducers. The culture was grown in the BSG medium (100 g/L BSG in water with 0.4 mM Cu^2+^ and 1 mM Tween 80) for 20 days at 20 °C.

The highest activity of extracellular laccase was achieved on the day 15 (215.39 U/L). It was about 65% higher compared to the enzymatic activity obtained for Sivakumar with the addition of inducers and ascorbic acid. Moreover, the activity at the level of approx. 110 U/L was observed even after 10 days (Figure 8).

It can therefore be concluded that the best waste material for the production of laccase by *K. bupleuri* G3 turned out to be the BSG. This raw material was also used to obtain *T. versicolor* ATCC 20869 laccase. The solid-state fermentation (SSF) process used BSG with a moisture content of 80–90% in the amount of 40–500 g. Laccase with an activity of 13,500 U/g of dry mass was obtained during 12 days of cultivation at 30 °C [34]. Apple pomace, which turned out to be completely ineffective in the case of *K. bupleuri* G3 culture, was used to obtain ligninolytic enzyme by *Phanerochaete chrysosporium* BKM-F-1767 [64], and other enzymes including xylanase *Aspergillus niger* NRRL 567 [65,66], pectinase *Bacillus pumilus* [67], cellulases *A. niger* [68] and *A. niger* NRRL 567 [65], and *Macrophomina phaseolina* [69]. Among other waste raw materials not tested in this study but used in the production of laccase, tomato pomace was utilized for cultivation of *Coriolus versicolor* [70].

### 3.6. Purification and Characterization of Laccase

#### 3.6.1. Purification of the Laccase G3 by Ammonium Sulphate Fractionation

To initially purify the laccase from the supernatant, salting out with ammonium sulphate was applied (Table 2).

Salting out with ammonium sulphate at 40% saturation of the supernatant allowed for the precipitation of about 21% of proteins with a total activity of 0.32 U. The proteins remaining in the supernatant were further precipitated with ammonium sulphate, reaching 60% saturation of the solution. This fraction showed the highest specific activity of laccase (0.5 U/mg), the highest degree of purification (1.67) and the highest process efficiency (58%). In the remaining fractions, the specific activity of the target enzyme was much lower (Table 2).

#### 3.6.2. SDS-PAGE

SDS-PAGE was performed for all concentrated protein fractions after salting out. The electropherogram (Figure 9) shows the protein band present in the post-culture supernatant and in one of the fractions for which the highest level of activity was achieved (60% saturation, lane 3 at Figure 9; the band is indicated by the red box). Such preliminary purification of the preparation allowed determining of the molecular weight of the tested protein. The laccase produced by *K. bupleuri* may have a mass of about 70–75 kDa (lane 3, Figure 9). For the remaining fractions, the potential laccase band of the desired molecular weight is very poorly visible, which results from a much smaller overall pool of proteins. Moreover, during the selection of the best waste material for laccase production, the analysis of SDS-PAGE electropherograms showed that the activity of laccase correlates with the presence of the protein band of about 70–75 kDa. In the supernatants where no laccase activity was observed, the band of this size was not identified (results not shown). The size of the G3 laccase is typical for fungal laccase. For example, *A. pullulans* NRRL 50381 produces a laccase of 60–70 kDa, while the enzyme from *A. pullulans* NRRL Y-2568 has a mass greater than 100 kDa [52]. Laccases produced by fungi from the genus *Trametes* also have a mass of 60–70 kDa; enzymes from *Populus euramericana* and *Myceliophthora thermophila* are slightly heavier—90 and 80 kDa, respectively [1].

#### 3.6.3. Properties of the Preliminary Purified *K. bupleuri* Laccase G3

The optimal conditions for the operation of the pre-purified extracellular laccase produced by the yeast *K. bupleuri* G3 were determined. Maximum activity in the 10 min reaction with ABTS was observed at temperatures in the range of 30–40 °C. It was noted that T_opt_ is not an absolute value as it decreases with increasing reaction time (Figure 10). A temperature of 30 °C was assumed to be the optimal temperature of action of G3 laccase. The study of the thermostability of this enzyme was carried out in the temperature range 4–80 °C during 30-, 60- and 120-min incubation. The temperature of 25 °C did not change the enzyme activity after 120 min. For a shorter time, i.e., 30 and 60 min, laccase remains stable at temperatures up to 35 °C, but above this value a very marked decrease in activity has already been observed. It was found that 30 min incubation of the enzyme at 40 °C causes a loss of 37% of the activity, while after 120 min only approx. 20% of the maximum activity remains. Moreover, the temperature of 50 °C causes a complete laccase inactivation during 30-min incubation.

The high thermolability of G3 laccase, its low optimum temperature (30 °C) and the fact that this enzyme retains over 60% of its maximum activity at 10 °C and approx. 40% at 0 °C, allow classifying of this biocatalyst to the group of enzymes adapted to the action at low temperatures, namely, psychrophilic enzymes. This is extremely important because only a few cold-adapted laccases have been described so far. Moreover, G3 laccase is the first enzyme of its kind for which activity was also observed at 0 and 4 °C. Most of the known multicopper oxidases are thermophilic enzymes characterized by high optimal operating temperatures, and the fungal laccases from the literature usually have T_opt_ of 50–60 °C [29,35,45,71,72,73,74]. Homologues with lower T_opt_ were also described, e.g., laccases from *A. pullulans* [51,52], *Monilinia fructicola* [39], *Aureobasidium melanogenum* [75], *V. volvacea* V14 [31] and *Cerrena* sp. HYB07 [72] operate at high activity at 30–45 °C, similarly to the *K. bupleuri* laccase reported in this study. Lower optimal temperature was reported for *Ganoderma lucidum* laccases (20–25 °C) [76], which additionally retains 65% of maximum activity in the reaction conducted at 10 °C. At the same time, however, it is characterized by greater thermostability than G3 laccase, maintaining approx. 90% activity after 4-h incubation at 40 °C. An interesting example of a laccase with a wide range of operating temperatures is the enzyme produced by *Pycnoporus* sp. SYBC-L1. It shows the highest activity at 70 °C, while at the same time during the reaction carried out at 0 °C and 10 °C it retains approx. 30% and 40% of the maximum activity, respectively. Interestingly, it also catalyzes the ABTS oxidation reaction at 100 °C, maintaining about 40% of the maximum activity [32]. Another example of a laccase with a low optimum temperature of action is the enzyme from *Cryptococcus albidus* (now *Naganishia albida*) described by Singhal et al. [15]. It is the most active at pH 2.5 and 20–30 °C. It has a relatively low thermal stability, with a half-life of 81 min at 25 °C, 77 min at 35 °C and 64 min at 45 °C [15]. Another extremely thermolabile enzyme is the laccase from *C. lagenarium*. Its optimal reaction temperature with ABTS is 40 °C, but after 1 h at 30 °C it loses 28% of its activity, and temperatures of 50–60 °C completely inactivate it within 10 min [16]. Finally, a homologous fungal enzyme, produced by *Trichoderma atroviride*, retains 100% and 80% of the maximum activity after 24-h incubation at 4 °C and 30 °C, respectively. Only above 50 °C, does 2-h incubation lead to a clear inactivation of the enzyme, which then retains approx. 20% of the maximum activity [28]. However, most of the reported laccases remain stable for at least 1 h at 50 °C [52,71,73,74,77,78].

The influence of pH on the activity of G3 laccase produced by *K. bupleuri* was determined. Maximum activity was observed at pH 3.5 for ABTS and at pH 7.0 for syringaldazine at 30 °C (Figure 11). This enzyme has a relatively high stability over a wide pH range. During 24-h incubation of the enzyme in solutions in the range of pH 3.5–8.5, it retains over 80% of its maximum activity, and in the narrower range of pH 4.5–6.0 as much as 95% of activity. In an environment with a pH below pH 3.5 and above pH 8.5, a sharp decrease in activity and inactivation of the enzyme was observed.

The optimum pH in reaction with ABTS observed for fungal laccases is most often in the range of 3.0–4.0 [16,28,31,35,45,72,75]. A slightly higher value, pH 5.0, was noted for *A. pullulans* laccases [51] and for *Thielavia* sp. [74]; and pH 4.5–5.0 for the laccases from *M. fructicola* [39]. A much more acidic operating environment is preferred by laccase produced by *Coprinopsis cinerea* Okayama 7, with its highest activity towards ABTS is recorded at pH 2.5 [71]. Values of optimal pH for syringaldazine oxidation are usually higher than observed for ABTS and are in the range of 5.5–7.0 [31,71,77,78]. The pH-stability of laccases also varies and depends on the manufacturer. Thus, laccases from *M. fructicola* [39], *C. lagenarium* [16], *T. hirsuta* [29] and *Thielavia* sp. [74] are stable in an acidic pH range of 3.0–5.0, whereas the laccase from *Trametes* sp. operates at alkaline pH 7.0–9.0 [35]. Broad pH-stability range was observed for several homologous laccases from *Chaetomium* sp. (pH 4.0–9.0) [73], *Elaeocarpus sylvestris* (pH 4.0–10.0) [32] and *Shiraia* sp. SUPER-H168 (pH 4.0–9.0) [78].

The apparent K_m_ and V_max_ values for ABTS were 0.584 mM and 1.14 U/mg at pH 3.5, respectively. Moreover, the enzyme from *K. bupleuri* G3 demonstrated a strong affinity to syringaldazine (K_m_ = 0.028 mM, V_max_ = 0.597 U/mg), which confirmed that the new enzyme is a laccase. Obtained K_m_ value for ABTS was lower than for the psychrophilic laccase from *C. albidus* (K_m_ = 0.8158 mM) [15] and higher than for *C. lagenarium* laccase (K_m_ = 0.34 mM) [16]. However, K_m_ value for syringaldazine was similar to values obtained for the laccase from *V. volvacea* V14 (K_m_ = 0.01 mM, V_max_ = 4.9 U/mg) [31] and the laccase from *C. cinerea* (K_m_ = 0.0226 mM) [71].

### 3.7. Application of Laccase in Synthetic Dye Decolorization Process

Laccase produced by *K. bupleuri* was used in the decolorization of methylene blue, alkaline fuchsin, crystal violet and Coomassie Brilliant Blue R-250 (Table 3).

The most popular synthetic dyes in the industry can be divided into azo, heterocyclic/polymeric or triphenylmethanes due to their chemical structure. *K. bupleuri* laccase can be successfully used in the process of removing dyes such as alkaline fuchsin and crystal violet. Both dyes are used in microbiology in many methods of staining prokaryotic and eukaryotic cells. In addition, alkaline fuchsin is used in molecular biology to stain glycoproteins and nucleic acids, e.g., with Schiff’s reagent. Crystal violet is used to dye fabrics. After 1 h of incubation of *K. bupleuri* laccase with the tested dyes at 30 °C, decolorization of approx. 32% and approx. 40% was observed for fuchsin and crystal violet, respectively. Crystal violet, also known as hexamethyl pararosaniline chloride, is a triphenylmethane dye, while alkaline fuchsin is not a specific dye but a mixture of triarylmethane dyes such as pararosaniline, rosaniline and new fuchsin. The differences in the degree of decolorization of these dyes prove that *K. bupleuri* laccase efficiently oxidizes only selected components of alkaline fuchsin with the structure of pararosaniline derivatives in the presence of a mediator. Moturi and Singara Charya studied the process of crystal violet decolorization by whole fungal cells of *Mucor mucedo*, *Polyporus elegans*, *T. versicolor* and *Lenzites betulina* [79]. The decolorization process observed during fungal growth ranged from 63 to 78% after 5 and 15 days of cultivation, respectively. The authors showed that several extracellular enzymes are responsible for this process, including laccases, manganese-dependent peroxidases and lignin peroxidases. Other studies indicated the possibility of using a purified laccase to effectively decolorize crystal violet. The laccase from *Pandoraea* sp. ISTKB had the ability to remove this dye in 82% during 1-h reaction with ABTS as a mediator, but using twice the concentration of the enzyme and a higher temperature than in the present study [80]. On the other hand, Tplac laccase from *T. pubescens* catalyzes the degradation of dyes without the addition of a mediator. The degree of decolorization of methylene blue was 25% and 17% by immobilized and free laccase, respectively, while for crystal violet it was 21% and 14%, respectively. The process was carried out at pH 5.0 at 50 °C with the use of an enzyme with an activity of 1 U/mL [81]. Bacterial laccase derived from the strain *Kurthia huakuii* LAM0618 also has the ability to degrade the dyes described in this study. In the reaction carried out at 60 °C for 1 h in the presence of 0.1 mM ABTS, approx. 35% degradation of alkaline fuchsin and approx. 60% degradation of crystal violet were observed. Without the addition of a mediator, the degree of discoloration was lower and amounted to approx. 25% and approx. 15%, respectively. The process of decolorization of methylene blue in the presence of ABTS was about 80% effective, and without the mediator, the dye remained intact [82].

Coomassie Brilliant Blue also belongs to the group of triphenylmethane dyes and is used in molecular biology to stain proteins. Forootanfar et al. investigated the effectiveness of three laccases from *Aspergillus oryzae*, *T. versicolor* and *Paraconiothyrium variabile* to decolorize this dye in a shaken culture at 35 °C [83]. Laccases with an activity of approx. 8 U/mL and hydroxybenzotriazole (HBT) as a mediator were used. The highest degree of decolorization was observed for *P. variabile* laccase; after 1-h reaction it was about 55%, and after 3 h as much as 90%. The *A. oryzae* and *T. versicolor* laccases showed lower efficacy, approx. 5% and 25%, respectively, after 1 h, and 13% and 30% after 3 h of incubation. In this study, laccase from *K. bupleuri* only achieved about 20% of the decolorization degree of Coomassie Brilliant Blue after 1 h of reaction, which corresponds to the commercial activity of *T. versicolor* laccase.

The last dye tested was methylene blue belonging to the group of heterocyclic dyes. It is a thiazine derivative used in medicine, chemistry and microbiology. The efficiency of decolorization of this dye by *K. bupleuri* laccase was similar to that of Coomassie Brilliant Blue and amounted to approx. 18%. Similar values were found for *P. variabile* laccase and it was about 15% discolored after 1 h of the reaction and 21% after 3 h. Commercial laccases from *A. oryzae* and *T. versicolor* show a much greater activity towards the removal of methylene blue, approximately 90% and 50% decolorization, respectively [83]. The degree of dye decolorization, similar to the studied *K. bupleuri* laccase, amounting to approx. 12%, was also found for laccase from *Pandoraea* sp. ISTKB, but the enzyme’s concentration was twice as high and the reaction was conducted at a higher temperature of 37 °C [80]. Lac2 laccase from *Cerrena* sp. HYB07 in the presence of acetosyringone (ACE) as a mediator showed a high decolorization efficiency of alkaline fuchsin, crystal violet and Coomassie Brilliant Blue, 81%, 87% and 100%, respectively, but using a 20-fold higher enzyme concentration. The degree of decolorization of methylene blue was clearly lower than that observed for the *K. bupleuri* laccase, only about 6% [72]. Similar results were obtained by Ademakinwa and Agboola [84] for *A. pullulans* laccases—4.5% decolorization of the dye in a 3-h reaction, whereas the laccase from *Shiraia* sp. SUPER-H168 described by Yang et al. [78] did not display any activity towards decolorization of methylene blue.

## 4. Conclusions

The conducted research reported a new laccase-producing strain—*K. bupleuri* G3 IBMiP. Enzyme production is induced by copper (II) ions and Tween 80 added to the medium after three days of the growth. Moreover, the positive effect of ascorbic acid on improving the quality of filtration and facilitating the purification of the enzyme by reducing the polymerization and degradation of carotenoid dyes produced by *K. bupleuri*, and so increasing the activity of laccase in the post-culture liquid, was demonstrated. The high activity of laccase G3 (215.4 U/L after 15 days) was obtained on a cheap substrate containing the BSG waste material. The new enzyme is characterized by cold adaptation manifested by high activity at low temperatures and low thermostability, which makes it an interesting biocatalyst in reactions carried out in cold environments. High thermolability allows for an easy inactivation of the enzyme. The potential of the enzyme in the processes of decolorization of synthetic dyes from the triphenylmethane dyes group and methylene blue belonging to heterocyclic/thiazine dyes was also reported. Further planned work on the new laccase from *K. bupleuri* G3 IBMiP includes the isolation and expression of the laccase-encoding gene in a heterologous host to shorten the cultivation time and increase the enzymatic activity.

It is worth emphasizing that so far, no enzymes produced by *K. bupleuri* yeast have been described in the literature. For the first time, this research shows the ability of these microorganisms to produce enzymes, including the cold-adapted laccase, which is extremely interesting from a biotechnological point of view.

## Figures and Tables

**Figure 1 biomolecules-11-00828-f001:**
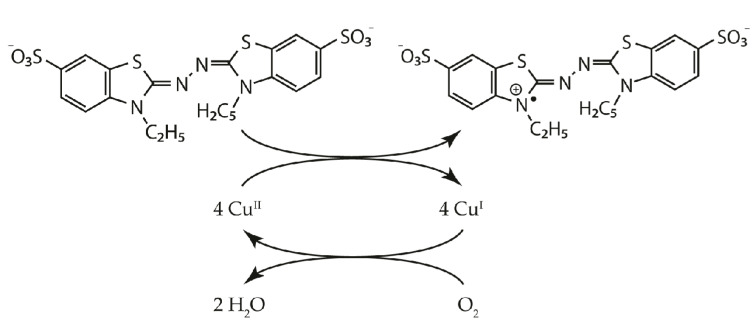
Scheme of a laccase catalytic cycle with 2,2′-azino-bis(3-ethylbenzothiazoline-6-sulfonic acid (ABTS) as a substrate, according to Riva, 2006 [8].

**Figure 2 biomolecules-11-00828-f002:**
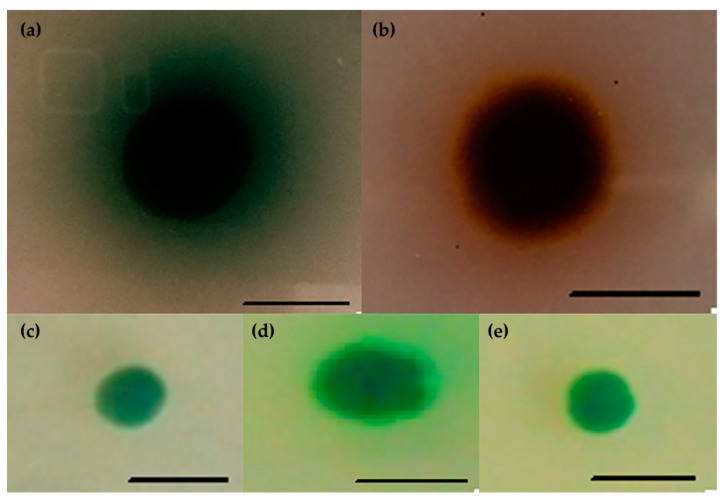
Screening for laccase synthesis. (**a**) Strain G3 IBMiP on medium with ABTS; (**b**) strain G3 IBMiP on medium with guaiacol; (**c**) *Leuconeurospora* sp. D59 IBMiP on medium with ABTS; (**d**) *Cryptococcus gastricus* DBVPG on medium with ABTS; (**e**) *Cryptococcus aerius* DBVPG on medium with ABTS. Black bars indicate the length of 10 mm.

**Figure 3 biomolecules-11-00828-f003:**
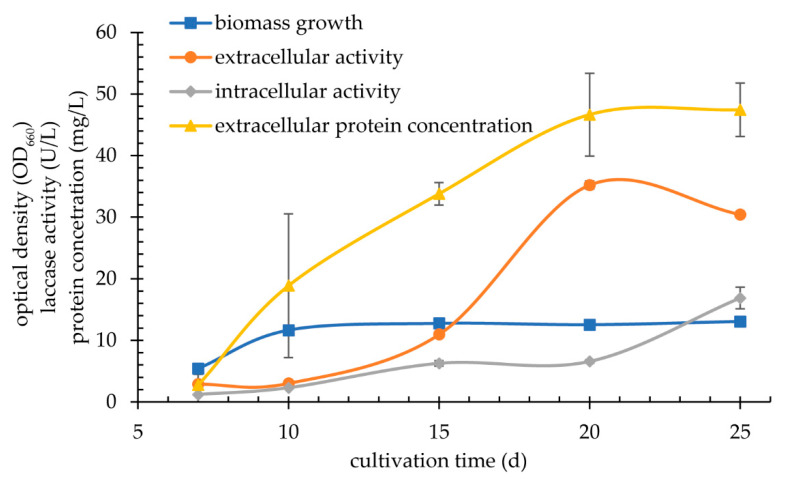
Growth, protein content, extracellular and intracellular laccase activity in the culture of strain G3 IBMiP in the Sivakumar medium.

**Figure 4 biomolecules-11-00828-f004:**
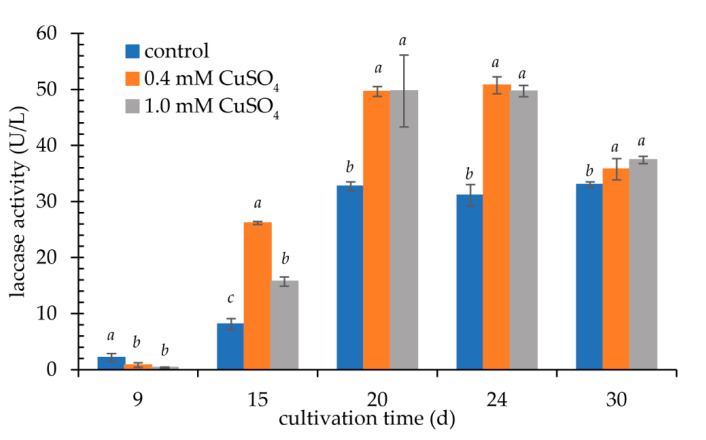
Influence of Cu^2+^ on laccase activity (cultivation in Sivakumar medium); the data presented as the means ± SD from at least three independent experiments; the means within single culture day that are marked with different letters (*a*–*c*) are significantly different.

**Figure 5 biomolecules-11-00828-f005:**
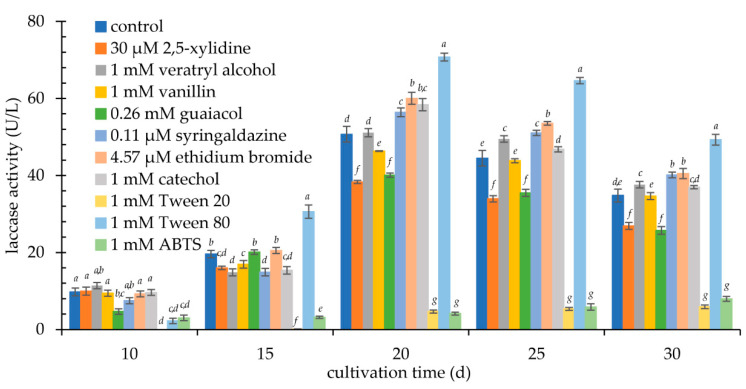
Influence of inducers (at the optimal concentration) on extracellular laccase activity (U/L) during 30-day culture; The data presented as the means ± SD from at least three independent experiments; the means within single culture day that are marked with different letters (*a*–*g*) are significantly different.

**Figure 6 biomolecules-11-00828-f006:**
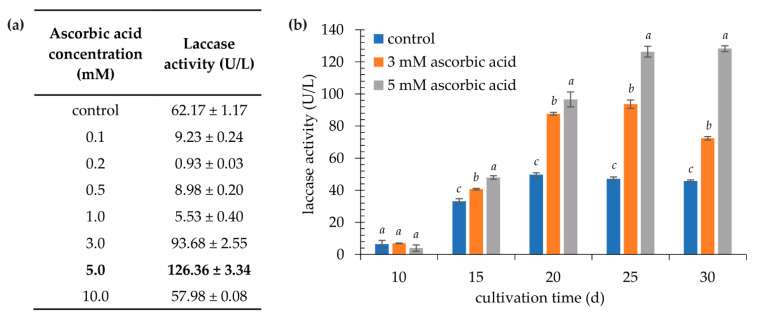
Influence of ascorbic acid on extracellular laccase activity. (**a**) Laccase activity after 25 days of cultivation with different concentration of ascorbic acid (the maximum activity with 5.0 mM of ascorbic acid is shown in bold); (**b**) laccase activity during cultivation in medium with 3 mM and 5 mM of ascorbic acid; the data presented as the means ± SD from at least three independent experiments; the means within single culture day that are marked with different letters (*a*–*c*) are significantly different.

**Figure 7 biomolecules-11-00828-f007:**
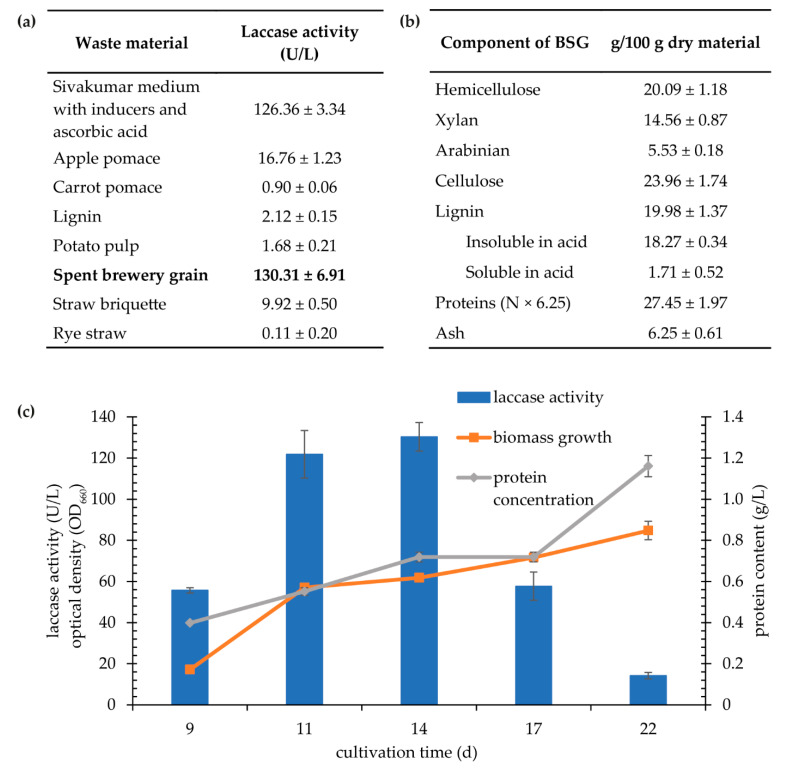
Cultivation of *K. bupleuri* G3 with waste materials. (**a**) Laccase activity after cultivation in Sivakumar media (without the presence of additional inducers) enriched with various waste products at a concentration of 100 g/L (the maximum activity with brewery spent SG) is shown in bold); (**b**) chemical composition of; (**c**) growth, protein content and extracellular laccase activity during cultivation in Sivakumar medium with 100 g/L of BSG.

**Figure 8 biomolecules-11-00828-f008:**
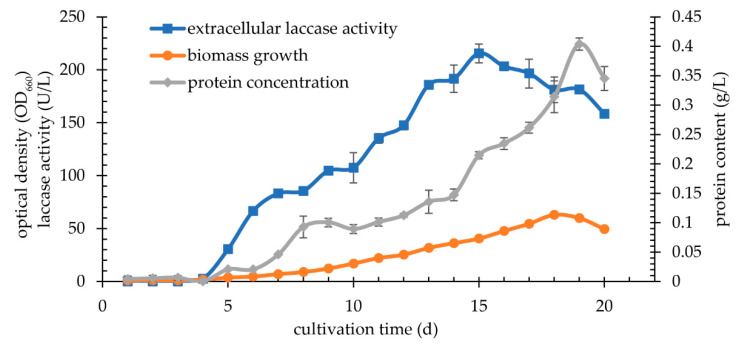
Kinetic of cultivation *K. bupleuri* G3 in BSG medium.

**Figure 9 biomolecules-11-00828-f009:**
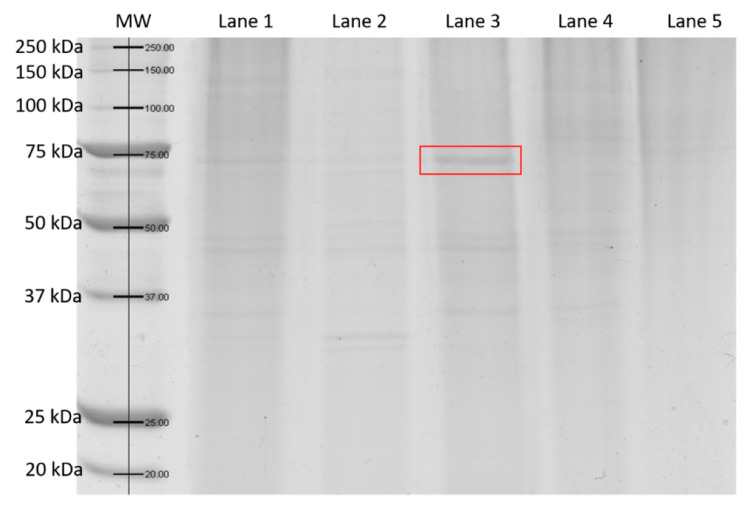
SDS-PAGE after salting out of laccase *K. bupleuri*; MW: molecular weight marker; Lane 1: crude post-culture supernatant; Lane 2: 40% saturation; Lane 3: 60% saturation; Lane 4: 80% saturation; Lane 5: supernatant after salting out with 80% saturation of ammonium sulphate. The laccase produced by *K. bupleuri* may have a mass of about 70–75 kDa (band at the lane 3 indicated using a red box)—the appearance of this band correlated with the highest specific activity of G3 laccase.

**Figure 10 biomolecules-11-00828-f010:**
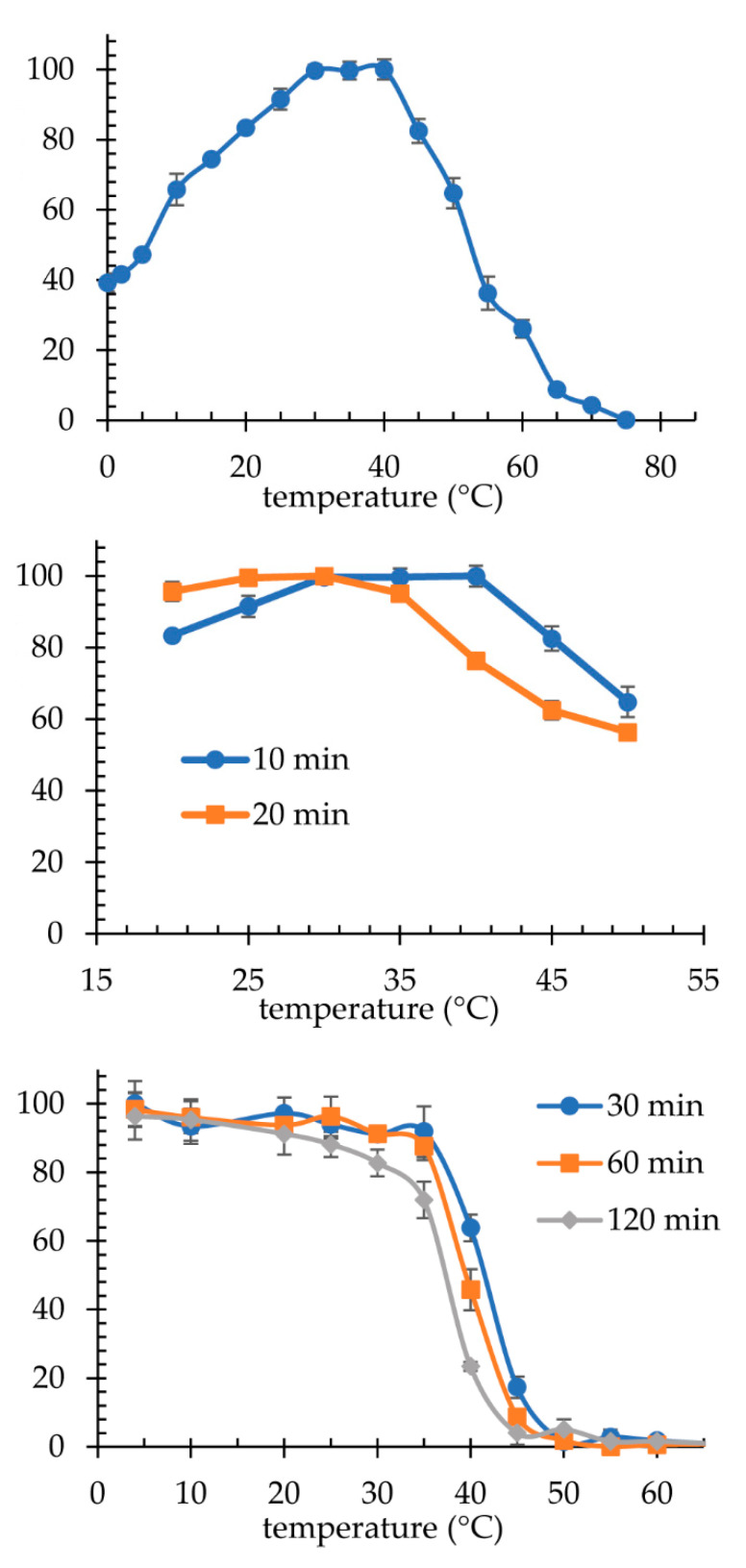
Properties of laccase *K. bupleuri*: optimal temperature (**upper panel**) and its variation for different reaction times (**middle panel**); thermostability during 30-, 60- and 120-min incubation (**lower panel**).

**Figure 11 biomolecules-11-00828-f011:**
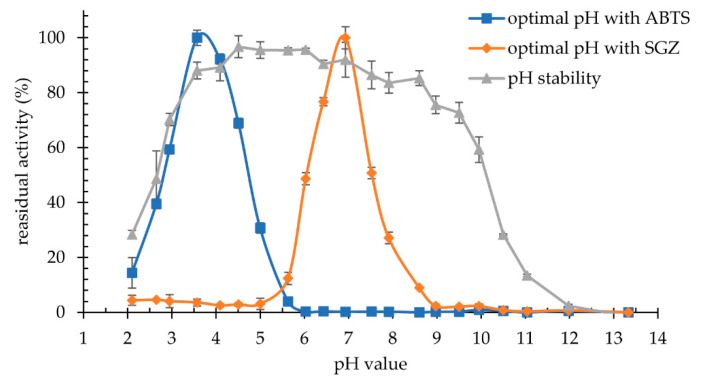
Optimal pH of reaction laccase *K. bupleuri* G3 with 2,2′-azino-bis(3-ethylbenzothiazoline-6-sulfonic acid) (ABTS) and syringaldazine (SGZ) and its pH-stability after 24-h incubation at 4 °C.

**Table 1 biomolecules-11-00828-t001:** Sequence of primers used for genetic identification of laccase producing strain.

Name of Primers	Sequence (5′ → 3′)
For amplification	
ITS5	GGAAGTAAAAGTCGTAACAAGG
LR6	CGCCAGTTCTGCTTACC
RLR3	GGTCCGTGTTTCAAGAC
V9	TGCGTTGATTACGTCCCTGC
For sequencing	
NL1FWD	GCATATCAATAAGCGGAGGAAAAG
NL4REV	GGTCCGTGTTTCAAGACGG
ITS1	TCCGTAGGTGAACCTGCGG
ITS4	TCCTCCGCTTATTGATATGC

**Table 2 biomolecules-11-00828-t002:** Fractionated salting out of laccase *K. bupleuri* G3.

Sample	Total Activity (U)	Total Protein (mg)	Specific Activity (U/mg)	Yield (%)	Purification (fold)
Crude post-culture supernatant	4.430	14.53	0.305	100%	-
40% saturation of ammonium sulphate	0.316	3.05	0.104	7%	0.34
60% saturation of ammonium sulphate	2.556	5.01	0.510	58%	1.67
80% saturation of ammonium sulphate	0.518	4.46	0.116	12%	0.38
Supernatant after salting out with 80% saturation	0.189	1.03	0.184	4%	0.60

**Table 3 biomolecules-11-00828-t003:** Decolorization of the synthetic dyes using laccase G3.

Dyes	Group	Chemical Structure	% Decolorization
Methylene blue	Heterocyclic/Thiazine dye	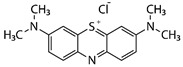	18.2 ± 2.5%
Alkaline fuchsin	Triphenylmethane dye and aniline dye	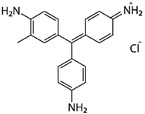	31.7 ± 3.3%
Crystal violet	Triphenylmethane dye	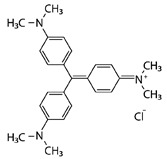	40.4 ± 7.0%
Coomassie Brilliant Blue R-250	Triphenylmethane dye	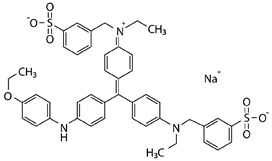	19.8 ± 5.8%

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
