# Peer review of "Screening of Novel Laccase Producers—Isolation and Characterization of Cold-Adapted Laccase from Kabatiella bupleuri G3 Capable of Synthetic Dye Decolorization"

_biomolecules, 2021, doi:10.3390/biom11060828_

Round 1

Reviewer 1 Report

In this work, the author obtained a laccase-producing strain by screening 41 cold-adapted strains of yeast and yeast-like fungi. It was identified as Kabatiella bupleuri G3 IBMiP according to the morphological characteristics and analysis of sequences. With ABTS as the substrate, the extracellular activity of laccase at the optimal pH 3.5 was 215 U/L after 15 days of growth in a medium with waste material and 126 U/L after 25 days of cultivation in a defined medium. They found that the production of laccase could be enhanced in the presence of copper (II) ions (0.4 mM), tween 80 (1.0 mM) and ascorbic acid (5.0 mM). They also confirmed that the optimum temperature for enzyme operation is in the range of 30-40°C and retains over 60% of the maximum activity at 10°C. The new laccase gives high thermolability and good degradation capability for the synthetic dyes when using ABTS as a mediator. They concluded that the K. bupleuri laccase could be used as a potential psychrozyme for environmental and industrial applications.

Furthermore, Km and Vmax values of laccase from Kabatiella bupleuri G3 IBMiP should be given and compared with the counterpart of cold-adapted laccase from Cryptococcus albidus and Colletotrichum lagenarium.

Reviewer 2 Report

The idea to search psychrophilic laccases is interesting and there is effectively potential applications. But I am not convinced by the protocol to prove the activity of psychrophilic laccase. A lot of peroxidases used also guaiacol and ABTS as substrate in the presence of hydrogen peroxide that could be produced by the yeast. So it is not a proof that it is laccases responsible for the bluish green or reddish brown color of the media. The authors used also a lot of substrates of laccase as inducers for the production and / or secretion of the enzymes and presented the results in Figures 4, 5 and 6. The authors compared the results between each conditions, but no statistics were made to know if the results are significantly different or not. In Figure 3, the authors discussed about laccase activity in the title of the Figure, but what is the proof that this is a laccase.

The authors prepurified and tried to isolate the enzyme responsible for ABTS activity. They showed a band on SDS PAGE electrophoresis around 70-75 kDa and concluded is it probably a laccase. The authors could have done a mass spectrometry analysis by cutting the band of this protein on the gel to have the peptide composition and identify clearly what is this protein. It is missing some key experiments to validate definitively the nature of this protein.

The authors also used ethidium bromide to induce the synthesis of laccases. The final aim is to detoxify the dye but the idea to use a DNA intercalant is very strange because it is certainly more toxic than the dyes themself.

Concerning the pH assays, it could have been interesting to use other substrate such as syringaldazine or 2,6-dimethoxyphenol to check the activity of the enzyme. ABTS is not a good choice to test the activity of laccase up to 6 units of pH.

Finally, the complete genome sequence of the strain Kabatiella bupleuri could be also obtained. It will permit to do bioinformatics and find new laccases.

Round 2

Reviewer 2 Report

The authors improved the manuscript. They tested the interference with peroxidase and prove there is no interference. So the activity tests are dependent of the presence or not of a laccase type enzymes. They performed activity assays with syringaldazine and showed now the enzyme is also active at pH 7. With the oldest version, it was not so evident. The authors also include results of statistical tests and Figures are modified in consequences. The SDS-PAGE gel is not well resolved even we can effectively see or guess a band around 70-75 kDa. Mass spectrometry analysis of this band could really convince definitively lectors that it is a laccase. It is really missing this experiment.  

Author Response

Reviewer 2 comment:

The authors improved the manuscript. They tested the interference with peroxidase and prove there is no interference. So the activity tests are dependent of the presence or not of a laccase type enzymes. They performed activity assays with syringaldazine and showed now the enzyme is also active at pH 7. With the oldest version, it was not so evident. The authors also include results of statistical tests and Figures are modified in consequences. The SDS-PAGE gel is not well resolved even we can effectively see or guess a band around 70-75 kDa. Mass spectrometry analysis of this band could really convince definitively lectors that it is a laccase. It is really missing this experiment. 

Reply:

We would like to thank the Reviewer for the acceptance of the introduced changes. We agree that the quality of the manuscript improved after the modifications suggested by the Reviewer.

Concerning the identification of the band of approx. 70-75 kDa as a laccase, we have indicated it on the Figure 9 and added a caption to the Figure (lanes 498-499: "(band at the lane 3 indicated using a red box) – the appearance of this band correlated with the highest specific activity of G3 laccase."). We have also added the explanation on how the appearance of this band correlates with laccase activity (lanes 478-479: “(60% saturation, lane 3 at Figure 9; the band is indicated by the red box)”, lanes 484-487: "Moreover, during the selection of the best waste material for laccase production, the analysis of SDS-PAGE electrophoregrams showed that the activity of laccase correlates with the presence of the protein band of about 70 - 75 kDa. In the supernatants where no laccase activity was observed, the band of this size was not identified (results not shown).").

We agree with the Reviewer’s comment that the mass spectroscopy (MS) is a convincing method to corroborate the identity of a protein. Psychrozymes are in a focus of interest in our research group since many years and in the past we performed the MS analysis, which gave ambiguous results for new proteins from psychrophilic organisms. Psychrozymes are often characterized by unique sequences that differ from their mesophilic, so far best described, homologues. Therefore, the peptides obtained during enzymatic digestion of cold-loving enzymes are not easy to identify on the basis of databases containing mostly the masses of peptides obtained after degradation of well-known and characterized proteins.

For this reason, as well as due to the potential of the reported G3 laccase, we decided that the most suitable method for us would be the search of the gene of G3 laccase in the native strain.

To this end,  we performed the sequencing of the genome of K. bupleuri G3 strain. In-depth bioinformatic analysis of the genome allowed for identification of three genes encoding for extracellular laccases of this psychrophilic yeast-like fungi. On the basis of characteristics of the target enzyme with a mass of approx. 70-75 kDa, one gene was selected, confirming the molecular weight of G3 laccase determined in this study. The sequence coding for the new laccase was now deposited in the GenBank database (@NCBI) and received an accession number MZ292708 (the record is being processed now). At present, the authors are preparing a manuscript of publication on the cloning and expression of the K. bupleuri laccase gene in an efficient expression system. It is worth mentioning that the native and recombinant enzymes have almost identical biochemical properties (Topt, pHopt, substrate specificity), therefore we were convinced that the band of 70-75 kDa which correlated with the laccase activity in the presented study could be identified as the G3 laccase.